# Cellular Protein Aggregates: Formation, Biological Effects, and Ways of Elimination

**DOI:** 10.3390/ijms24108593

**Published:** 2023-05-11

**Authors:** Jun-Hao Wen, Xiang-Hong He, Ze-Sen Feng, Dong-Yi Li, Ji-Xin Tang, Hua-Feng Liu

**Affiliations:** Guangdong Provincial Key Laboratory of Autophagy and Major Chronic Non-Communicable Diseases, Institute of Nephrology, Affiliated Hospital of Guangdong Medical University, Zhanjiang 524001, China

**Keywords:** protein aggregates, proteostasis, aging, age-related neurodegenerative diseases, autophagy, ubiquitin-proteasome system

## Abstract

The accumulation of protein aggregates is the hallmark of many neurodegenerative diseases. The dysregulation of protein homeostasis (or proteostasis) caused by acute proteotoxic stresses or chronic expression of mutant proteins can lead to protein aggregation. Protein aggregates can interfere with a variety of cellular biological processes and consume factors essential for maintaining proteostasis, leading to a further imbalance of proteostasis and further accumulation of protein aggregates, creating a vicious cycle that ultimately leads to aging and the progression of age-related neurodegenerative diseases. Over the long course of evolution, eukaryotic cells have evolved a variety of mechanisms to rescue or eliminate aggregated proteins. Here, we will briefly review the composition and causes of protein aggregation in mammalian cells, systematically summarize the role of protein aggregates in the organisms, and further highlight some of the clearance mechanisms of protein aggregates. Finally, we will discuss potential therapeutic strategies that target protein aggregates in the treatment of aging and age-related neurodegenerative diseases.

## 1. Introduction

As the material basis of vita, proteins play essential roles in various forms of life activities in organisms. Therefore, it is important to maintain the correct conformation of protein in mammalian cells. Mammalian cells have developed a system called proteostasis network to maintain more than 10,000 different proteins’ correct conformation [1]. In healthy mammalian cells, a proteostasis network contains molecular chaperones and proteolytic machinery and their regulators, which coordinate with each other to ensure the maintenance of proteostasis [1]. However, it is a challenging task to maintain proteostasis, especially in the face of various external and endogenous stresses that accumulated in aging. These stresses can result in a decrease in proteostasis network capacity and proteome integrity, causing the accumulation of misfolded and aggregated proteins, which will specifically affect the postmitotic cell types such as neurons [2,3]. As the symbol of proteostasis imbalance, protein aggregation can be found in aging cells and tissues and damaged organs [4,5]. Moreover, recent findings establish a central role of enhanced proteostasis to prevent the aging of somatic stem cells in adult organisms. Notably, proteostasis is also required for the biological purpose of adult germline stem cells, which are to be passed from one generation to the next. Beyond these links between proteostasis and stem cell function, these insights demonstrate that embryonic stem cells and induced pluripotent stem cells exhibit an endogenous proteostasis network that not only delays senescence but also maintains the ability to reproduce. Besides the essential roles of the proteostasis network in postmitotic neurons, it also play important roles in maintaining stem cell function [6], such as modulating stem cells pluripotency and differentiation as well as suppressing aggregation of disease-related proteins [7].

As the hallmark of aging, loss of proteostasis can be found in many age-related diseases and degenerative diseases, such as Alzheimer’s disease (AD), Parkinson’s disease (PD), Huntington’s disease (HD), and idiopathic cardiomyopathy [8,9,10,11,12,13]. Protein aggregates caused by loss of proteostasis can lead to cell dysfunction via damaging the lysosomes, inducing endoplasmic reticulum stress (ER-stress), causing DNA damage, and perturbating Ca^2+^, all of which can promote the progression of age-related diseases [14].

In this review, we will briefly summarize the composition and causes of protein aggregates in mammalian cells, and then we will systematically introduce the function of protein aggregates in mammalian cells, and further highlight some of ways to the clear the protein aggregates. Finally, we will discuss potential therapeutic strategies that target protein aggregates for the treatment of aging and age-related neurodegenerative diseases.

## 2. The Components of Protein Aggregates

Protein aggregates can be formed from almost all kinds of proteins in cells. The state of protein aggregates covers liquid (monomer), liquid-like or solid-like (oligomers), and solid (insoluble aggregation) [15]. In aging *C. elegans*, though the most abundant proteins were 10 times more soluble than the least abundant proteins, the most abundant proteins contribute most to the total aggregate load [16]. Apparently, the high solubility of abundant proteins is insufficient to protect them from age-dependent aggregation. Interestingly, in long-lived daf-16 mutant *C. elegans*, protein aggregates were accumulated significantly more than that of age-matched WT animals [16]. This phenomenon suggests that long-lived daf-16 mutant worms can handle protein aggregates well, driving aberrant, potentially toxic proteins into aggregates. Packing up damaged abundant proteins and separating them in order to prevent proteostasis imbalance can help daf-16 mutant worms live longer.

In the organism proteome, most proteins have special segments that are not likely to form a defined 3D structure which is characterized by their biased amino acid composition and low sequence complexity. These structures are known as intrinsically disordered proteins (IDPs) and intrinsically disordered protein regions (IDPRs) [17,18]. It may explain why protein aggregates are mainly formed by the most abundant proteins. Because the most abundant proteins contain IDPRs, when facing stress environments, they tend to perform liquid-liquid phase separation (LLPS) and turn into protein aggregates if stress factors persist [19]. In most cases, protein phase separation is closely linked with the presence of IDRs and IDPRs in the phase-separated proteins [20]. However, IDPs and IDPRs do play a key role in life activities such as protein modification and cellular signaling and regulation [21]. IDPs and IDPRs are the key structures to regulate cellular signals, as these motifs combine varieties of molecules to cascade or terminate signals, and they also promote flexible sites for post-translational modification to assist the activity of the kinase. Also, IDPs and IDPRs can regulate the amount of kinase by forming LLPS in order to trap some kinases and restrict those activities [18].

Besides the most abundant proteins, protein aggregates also contain other matter. Nucleic acids, such as RNA, can intertwine proteins, aggravating protein aggregate accumulation [22,23]. Eukaryotic cells have numerous dynamic membrane-less organelles, called RNP granules, such as nucleolus, Cajal bodies, stress granules, and P-bodies. These organelles can bind with RNA to perform physiological functions. In addition, aberrant regulation of RNA-RNP combination could explain the reason for the pathological stress granules formation in certain diseases [24,25]. In protein aggregates, RNAs are likely to act as ‘scaffolds’, which string misfolded proteins together [25,26,27]. RNA-dependent protein aggregates were widely found in degenerative diseases. In amyotrophic lateral sclerosis (ALS) and frontotemporal dementia (FTD), misfolded TDP-43 protein did not perform the function of protecting mRNA, rather than wrap around with mRNA and other proteins [28]. This process accelerates the formation of protein aggregates and makes removal more difficult. Once proteostasis is damaged, proteins and RNA can wrap together and change the stability of LLPS, promoting solid-like aggregates accumulation [25]. However, as the longest cells in the body with the most longevity, neurons use LLPS to create separate compartments for RNA and protein in order to transport matter and keep biochemical reaction accurately conducted [27]. This phenomenon could explain why neurons tend to form protein aggregates which cause cell dysfunction.

A relatively simple composition of protein aggregates also causes diseases. In AD patients’ neurons, β-amyloid 1–42 (Aβ 1–42) and tau protein aggregates are the most significant factors causing neurodegeneration [29,30,31]. Lewy bodies, which are accumulated by misfolded α-synuclein (α-syn), can be found in PD patients’ substantia nigra. They disrupt normal cellular architecture so that cell function is damaged, eventually leading to cell death [9,32,33]. Abnormal accumulation of Huntingtin (Htt) protein in the corpus striatum causes disordered motor movements, personality changes, and premature death in patients [10]. In cardiovascular diseases, protein aggregates also play a key role in the initiation process of diseases. In mouse models, αB-crystallin (CryAB) aggregates in cardiomyocytes can cause cardiac failure when the mouse is still at a young age [34]. High-temperature requirement protein A1 (HTRA1) aggregations and HTRA1 substrates accumulation are the key links for cerebral autosomal dominant arteriopathy with subcortical infarcts and leukoencephalopathy (CADASIL) [35]. How to clear protein aggregates is the key strategy to resist some hereditary cardiomyopathies. Not only do permanent cells such as neurons and myocardial cells face protein aggregation challenges, but highly active transporting epithelia such as renal tubules are also at risk of protein aggregates due to high protein turnover and/or oxidative stress. When stress load increases, such as aldosterone stimulation, renal tubule cells keratin 5, the ribosomal protein RPL27, ataxin-3, and even HSPs would be abnormally found accumulating in protein aggregates [36]. The influence of protein aggregates in renal tubule cells on renal function has still not been understood so far.

## 3. Protein Aggregation Caused by Losing of Proteostasis

A proteostasis network is required for mammalian cells to keep protein quality control, reduce misfolded protein, and maintain proteostasis [37], but as time went by, the ability of mammalian cells to keep proteostasis declined with aging. There are many reasons for organism proteostasis imbalance, which can be attributed to four points: first, there are not enough molecular chaperones in the cytoplasm to help proteins correctly fold; second, cells cannot appropriately react to endoplasmic reticulum stress (ER stress); third, digestion of damaged proteins is blocked for lysosomes degradation; fourth, protein aggregates accumulate due to above reasons [37]. Therefore, protein aggregate is the outcome event of proteostasis imbalance in the proteome, and it is the connector that leads to cell structure and function damage, aggravating organ failure.

The proteostasis network can maintain protein quality by correctly guiding polypeptide chain folding, maintaining the structure of proteins, and digesting damaged proteins [2]. However, in normal cells, some common proteins are close to their solubility limits [38,39] or are indeed supersaturated [16,40], which makes them stay in metastable ‘subproteome’. When the intracellular environment is perturbated, the metastable ‘subproteome’ aggregates into insoluble proteins, interfering with vital movement. In order to handle these insoluble proteins, more molecular chaperones could combine with them, this process aggravating polypeptide chain misfolding for lack of available molecular chaperones. If unfold protein response (UPR) was not activated in a timely manner, the cells would be caught in a vicious circle of proteostasis imbalance. However, protein aggregating also interacts with proteostasis, playing a special role in maintaining proteostasis. Intracellular environment perturbating can cause LLPS and even accumulation of protein aggregates. The intermediates process, LLPS, can also maintain proteostasis in some special ways. When misfolded proteins are in LLPS state, this special state equally limits them to a designated space in the cell. Moreover, proteasomes and related marker proteins such as p62/SQSTM1 also recruit into LLPS. This process improves the efficiency of proteasomes and autolysosomes to clear misfolded proteins [41,42].

Protein aggregates, such as amyloid deposition, can be found from Alzheimer’s disease to type II diabetes, most of which are typical age-related diseases [5]. Protein aggregates are particularly prominent in the pathogenesis of neurological diseases [30], such as Aβ 1–42 for Alzheimer’s disease [29], tau aggregation for Alzheimer’s disease [31],α-syn for Parkinson’s disease [9], and Rosenthal fibers for Alexander disease [43]. In molecular structure, amyloid deposition and many protein aggregates were characterized by β-sheet, and hydrogen bonding is the key structure to maintaining β-sheet [5]. Due to hydrogen bonding and β-sheet, the amyloid state has a lower free energy (G) than the native state (ΔG < 0), which means that the protein native state is kinetically metastable [38,44]. Once the intracellular environment fluctuates, such as heat stress and oxidative stress [45], the protein’s native state would spontaneously transform into an amyloid state from receiving enough activation energy to get over the energy barrier. As a molecular structural property of proteins, too many free radicals in the cytoplasm may accelerate the native protein’s transformation into the amyloid state in aging cells.

Staying in metastable ‘subproteome’, many proteins would trend to LLPS and even aggregation. However, this process is reversible in most situations when stress disappeared. However, some mutations in RNA-binding proteins (RBPs) can accelerate the process of LLPS and turn RBPs from reversible fibrils to irreversible fibrils [46,47]. In many neurodegenerative diseases, LLPS is the primary process of forming amyloid proteins such as α-syn, FUS, tau, and TDP-43, all of which can aggregate and form into amyloid proteins through irreversible LLPS and damage to neurons [48,49].

## 4. The Harm of Protein Aggregates

Protein aggregates are the normal matter of cellular metabolism, all cells must handle protein aggregates, however, if the cells could not eliminate protein aggregates below the threshold, they would damage cells in many ways and accelerate cell aging [14] (Figure 1). The ways that protein aggregates damage cells are multitudinous, weaving together with mutual promotion, becoming a complex network. For example, protein aggregates disrupt the membrane system, causing an imbalance of calcium homeostasis and lysosomal function incompetence. ROS, a by-product of protein aggregates, not only destroys DNA, but also disrupts the protein synthesis environment, which causes more serious consequences such as protein synthesis error and aggregates accumulation, creating a vicious circle. Therefore, stopping the interference of protein aggregates and breaking the chain of destructive effects is the key target of treatment for protein aggregate accumulation.

### 4.1. Interference with Lysosomal Function

Protein aggregate accumulations can be partially attributed to lysosome dysfunction. Correspondingly, lysosome dysfunction is also the phenotypic of protein aggregates accumulation [50,51]. The current study shows that protein aggregates interfere with lysosome via directly damaging lysosome structure and function, or disturbing lysosomal-associated genes expression and fusion to indirectly reduce degradation capacity. In Alzheimer’s disease models, though Aβ 1–42 aggregations were wrapped by autolysosomes, they still stayed in autolysosomes and even destroyed the integrity of autolysosomes, which caused leakage of hydrolase and broader cellular dysfunction [50]. However, improving the activity of autophagy could delay the onset of neurodegenerative disorders because it would reduce protein aggregate accumulation [52]. In addition, we should pay attention to another phenomenon: protein aggregates interfere with lysosomal gene expression. Also in Alzheimer’s disease, tau proteins accumulate into aggregates, which is another hallmark pathology. Abnormal tau accumulation could inhibit the transcription of IST1 expression, the key factor of autophagosome formation, via activating the CEBPB-ANP32A-INHAT pathway [51]. Upregulating IST1 or downregulating ANP32A can break the vicious cycle and reduce protein aggregates. Besides inhibiting gene expression, tau aggregations could also accelerate microtubule disassembly, inducing a massive buildup of autophagosomes in neuronal processes [53,54]. Therefore, tau aggregations accumulation can also impair autolysosome formation by disrupting microtubule dynamics and axonal transport. Analogously, in Parkinson’s Disease, α-syn protein aggregating also harms the degradations of autolysosomes [55]. The α-syn protein not only disrupts microglial autophagy initiation via Tlr4-dependent p38 and Akt-mTOR signaling in substantia nigra, but also piles up in autolysosomes to hamper them, degrading metabolic wastes [55]. This adverse outcome accelerates the apoptosis of microglia cells and inflammatory infiltration of the substantia nigra and drives the progression of Parkinson’s disease. We summarize that protein aggregates interfere with lysosome function via two pathways: first, directly damaging the lysosome structure and function which cause metabolic waste accumulating in cytoplasmic and leakage of hydrolases; and second, disturbing lysosomal associated genes expression or fusion of lysosomes to indirectly reduce degradation capacity. Both of these pathways break the intracellular environment, interweaving with each other and causing a more complex mechanism of injury.

### 4.2. Disruption of the Protein Synthesis Environment

Misfolded proteins are the key elements of protein aggregates, which means that any reason for protein synthesis interference could exacerbate protein aggregate accumulation [56]. Similarly, protein aggregates can disrupt the protein synthesis environment via induction of ER stress. In PD patient midbrain cultures, α-synuclein was aggregated, the aggregates induce ER fragmentation and compromise ER protein folding capacity, leading to protein misfolding and aggregation [57]. In the amyotrophic lateral sclerosis (ALS) mouse model, the aggregates-related protein C9orf72 can cause ER stress response [58]. Therefore, reducing ER stress is also a potential target for weakening protein aggregates’ influence. Many studies prove that limiting ER stress and activating UPS can relieve the negative impact of protein aggregates [56,59]. Some researchers suggest that ROS may be the key mediator between protein aggregates and ER stress [60,61]. However, we still do not comprehensively know how protein aggregates destroy the protein synthesis environment.

### 4.3. Induction of DNA Damage

DNA damage is a key symbol of cell aging and also a typical outcome of protein aggregate accumulation [62]. We notice that in many neurodegenerative diseases characterized by protein aggregates, neurons are observed DNA damage [29,63,64,65]. Focusing on DNA injury mechanism, protein aggregates damage DNA via three main modes: (1) Touching DNA and directly destroying its structure. (2) Injuring DNA indirectly through mediums such as ROS. (3) Interfering with the DNA repair system, causing an accumulation of damages [62]. These modes are not separately fought; they intertwine with each other and mutually reinforce. A. Suram et al. research showed that Aβ 1–42 has DNA-nicking activity similar to nuclease [21]. Further studies revealed that Aβ 1–42 causes open circular and linear forms in supercoiled DNA and also clearly evidenced the physical association of protein-DNA complex via transmission electron microscopy (TEM). The broken DNA was repaired via homology recombination which may generate frameshift mutation. Meanwhile, in the α-syn aggregation cell model, ROS accumulation was observed because of α-syn aggregation induction. ROS is the key poison that can damage DNA structure. V. Vasquez et al. observed that α-syn overexpression and oxidative stress significantly enhanced DNA damage in the neuronal genome [64]. While protein aggregates can injure DNA, the DNA damage repair system executes the task of protecting and repairing fragile genetic material. However, protein aggregates also interfere with this life-saving straw. Lior Weissman et al. found that AD patients’ brain has base excision repair (BER) deficiencies, suggesting a decreased capacity to repair oxidative DNA damage [65]. Tyler Fortuna et al. attempted to relieve the symptom of protein aggregates accumulation in neurons by improving DNA damage repair ability. And the result shows that superior DNA repair abilities suppress protein aggregates-mediated neuropathogenesis and toxicity in vivo [63].

### 4.4. Imbalance of Calcium Homeostasis

Calcium is the key ion of life activities, performing as an electrical signal carrier, a messenger, and the catalytically active center of many enzymes. Cells have an ingenious mechanism to maintain calcium concentration. However, protein aggregates can destroy the balance of calcium exchange, causing more disorder in biochemical metabolism [12,61,66]. Some researchers have discovered that certain types of protein aggregates could insert into membrane structures and play the role of a non-selective ion channel [66,67,68,69]. For example, Aβ 1–42, a typical protein aggregation, can form a pore transmembrane structure with an 8–25 nm outer diameter and 2–6 nm inner diameter [68,69]. Those aggregates channels can appear in the cytomembrane, changing the resting potential, but can also appear in the smooth endoplasmic reticulum, the largest calcium store in cells. When a pore transmembrane structure is formed in smooth endoplasmic reticulum, it means that lots of calcium ions have been lost to the cytoplasm, causing a disorder in metabolism. Furthermore, protein aggregates can increase membrane conductance and permeability to charged species by spreading the lipid head groups apart, consequently thinning the bilayer and lowering the permeability barrier, causing calcium ions to transmembrane flow [70,71,72]. To maintain calcium homeostasis, the calcium pump must keep working, which wastes precious energy in the neuron. On the other hand, calcium homeostasis is more similar to a result of the contents disclosure of membrane system disruption. Interestingly, protein aggregates are seen not to change calcium homeostasis via activating calcium channel protein. This is because even when calcium channels are blocked, the effect of calcium imbalance still existed [67]. Limiting the disruption of calcium homeostasis by protein aggregates is the key tool to preventing cellular senescence and degeneration.

### 4.5. Disrupts the Membrane System and Production of ROS

The membrane system separates different cell biochemical reactions and provides a smooth environment for metabolic processes. The interaction of protein aggregates with lipid membranes has been widely reported [73,74,75,76]. This primarily occurs through a physical mode of punching holes in the membrane and even emulsifying lipid bilayers. Entering into the membrane and creating transmembrane pores is one mode that not only causes membrane disruption, but also calcium imbalance [67,68]. Alternatively, a carpeting effect of Aβ 1–42 has been proposed, which is thought to result in a general increase in membrane conductance either by membrane thinning or a lateral spreading of lipid headgroups [70,71,72]. Even protein aggregates such as surfactants can disrupt the integrity of the membrane via emulsifying lipid bilayers [69,77]. Terminating the impact of protein aggregates on cellular membranes is the target of potential treatment.

ROS is the by-product of protein aggregates, which can attack any substance in the cell due to its strong oxidation [60,61,66,78]. In general, it is not clear why protein aggregates or their precursors trigger ROS production. Various hypotheses have been put forward, including an increase in oxidative metabolism to clear the excess of free Ca^2+^, the impairment of the functionality of the ER, and mitochondria ROS production [79]. Reacting with transition element ions, such as Fe^2+^ and Cu^2+^, protein aggregates subsequently converted to hydroxyl radicals. However, hydroxyl radical formation was inhibited by the inclusion of catalase or metal chelators [79].

## 5. The Way to Eliminate Protein Aggregates

Because protein aggregates are vital in influencing proteostasis imbalance, organisms developed many ways to protect themselves from protein aggregates. The methods of resisting protein aggregates can be summarized as following points: (1) Refolding misfolded proteins and depolymerizing aggregates. (2) Degrading protein aggregates via ubiquitin-proteasome pathway (UPP) or autophagy-lysosome pathway (ALP). (3) Alleviate protein aggregates load via asymmetric cell division (ACD) or exocytosis [80,81]. All of these are protecting proteostasis balance.

Refolding and depolymerizing protein aggregates are the first choice of cells to maintain proteostasis balance in a normal state for saving precious resources [23]. Though cells are still healthy, protein aggregates can be produced for intracellular environment fluctuations. Recycling or rescuing misfolded proteins in an efficient and economical way is of particular importance. Molecular chaperones can perform this task well [2,23]. The heat shock protein family (HSPs) is the most typical molecular chaperone protecting proteostasis. ATP-dependent HSP70 is the most important protein that can depolymerize misfolded proteins with the assistance of HSP110 (Figure 2A). In addition to HSP70, small heat shock proteins (sHSP) also play a role in repairing misfolded proteins by actively gathering them, separating damaged proteins into small protein aggregates, and preventing further misfolding. Interestingly, besides HPS, ubiquitin, the tag protein of protein aggregates, can also execute the mission of depolymerizing protein aggregates [22]. Ubiquitin can break the structure of the C-terminal ubiquitin-associating domain (UAB), a key construction of protein oligomerization and LLPS. Via combining polyubiquitylation in UAB, protein aggregates in LLPS can depolymerize again and return to the soluble state. This is precisely because refolding and depolymerizing protein aggregates requires many molecular chaperones. When facing stress, cells could be confronted with a dilemma, which is the lack of available molecular chaperones if ER-stress response cannot activate and produce enough molecular chaperones [82]. ER-stress response deficiency causes insufficient synthesis of molecular chaperones. Therefore, there is a complex interaction network between protein aggregating and proteostasis.

The ubiquitin-proteasome pathway (UPP) or autophagy-lysosome pathway (ALP) are the key systems of protein quality control in cells [83] (Figure 2B). Both of them are degraded misfolded proteins, but they handle different types of misfolded proteins. Because of the limitation of the narrow channel of the proteasome, UPS primarily deals with soluble misfolded proteins and unfolded polypeptides. When facing the gathering of misfolded proteins and even protein aggregates, UPP is helpless. To eliminate these damaged proteins, ALP was applied to wrap up protein aggregates and combine with the lysosome for enzymolysis [84,85]. To label the protein aggregates and target them to UPP and ALP, ubiquitin (Ub) is necessary for marking protein aggregates. Protein aggregates can be identified and conjugated ubiquitin by a hierarchically acting enzymatic cascade. Via Ub-conjugating enzyme linking Ub thioester, activated Ub, to substrate specificity [84,86]. When substrates are progressively modified with Ub, either at the N terminus (Met1) or at a lysine side chain of Ub, various linear or branched Ub chains are built. Poly-Ub link as a potent signal, recruiting intrinsic Ub receptors of the proteasome (Rpn10 or Rpn13) or shuttle factors that are equipped with both a Ub-binding domain and a domain that binds to the proteasome and guides ubiquitinated protein to the proteasome. When aggregates surpass the processing limitation of the proteasome, Ub chains can combine with autophagy receptors such as P62, NBR1, and TAX1BP1 [87]. After that, those receptors induce autophagosome, enveloping the protein aggregates and fusing with the lysosome to digest them.

For resting cells or dividing cells, asymmetric cell division (ACD) is also a good way to reduce protein aggregates in daughter cells [80,81,88] (Figure 2C). G0 cells such as mature neurons or myocardial cells cannot reduce protein aggregates via ACD, which may explain why the nervous system and cardiovascular system are sensitive to protein aggregates. However, they can secrete aggregates via exocytosis as well. Interestingly, in neuron development, ACD is the main way to protect daughter cells from protein aggregates [88]. In embryonic *Drosophila* neuroblast, protein aggregates were transported to the microtubule organizing center (MTOC) and interacted with the peri-centriolar material (PCM) by dynein [89]. When centrosomes separated from each other during mitosis, the mother centrosome (older centrosome) would drag the PCM to one side of the cell. In this process, protein aggregates can also be enriched to one side of the spindle body. The daughter centrosome would form the new PCM with fewer protein aggregates [90,91,92]. Finally, the daughter cell, which inherits the protein aggregates, could undergo apoptosis, while the other cell would survive for obtaining better resources [93,94]. A mature cell that has lost the ability to divide can also discard protein aggregates through extruded membrane-surrounded vesicles called “exophers” that can harbor protein aggregates and organelles [95,96] (Figure 2D). Similar to ACD, the thin thread-like tube can induce protein aggregates accumulating on one side of the neuron, and then form a compartment, which is an average of 3.8 μm large, the same as a neuron. Exophers ultimately disconnect from the originating neuron [95]. This process is similar to mitosis, other than the nucleus, mother cells are completely retained. This pattern to remove protein aggregates may be the supplement measure for permanent cells for losing the ability of mitosis [96].

Though there are no pieces of evidence revealing that cells can dismantle mature protein aggregates and amyloid fibrils into soluble proteins through the molecular chaperone pathway, some chemical chaperones were found to destabilize or disaggregate misfolded or aggregated states of polypeptide chains by regulating the viscosity, melting point, and ionic strength of biological fluids. For example, tauroursodeoxycholic acid (TUDCA) supplementation can prevent cognitive impairment and amyloid deposition in APP/PS1 mice [97]. Besides bile acids, trehalose and betaine can also disrupt protein aggregate into protein soluble assemblies by changing ionic strength and stabilizing hydrophobic amino acids from hydrophobic protein–protein interactions [98,99]. Catechin derivatives, non-steroidal anti-inflammatory drugs (NSAIDs), anthracycline, and tetracycline derivatives also show the ability to disassemble protein aggregate. However, they still need more research to clarify the mechanism and reveal potential risks before use in clinical treatment [100,101,102,103].

## 6. The Function of Protein Aggregates in Development of Aging-Related Diseases

Much evidence discovered that protein aggregates have a close relationship with aging-related diseases. Limiting protein aggregates can be a useful method to change the development of diseases. From neurodegenerative diseases to diabetes and idiopathic cardiomyopathy, protein aggregation is the key factor that promotes disease development. Inhibiting the formation of protein aggregation or promoting the elimination of protein aggregation also show a good effect on aging related diseases.

Neurodegenerative diseases are the best-known diseases caused by protein aggregates, so they have been intensively studied. People proposed many hypotheses to explain the reason protein aggregates cause neurodegenerative diseases. Alzheimer’s disease, Parkinson’s disease, Huntington’s disease, and Alexander’s disease are all found to be linked to protein aggregates [9,29,31,43]. Though they accumulated different kinds of protein aggregates in different locations of the brain, they do lead to dysfunction, atrophy even, and apoptosis in the corresponding parts [9,32,33]. These abnormalities of cells and tissues present with all the familiar neural symptoms we know. Current studies have suggested that clearing protein aggregates or delaying protein aggregates accumulation can effectively slow down the progression of symptoms. Therefore, clearing protein aggregates is expected to become a treatment for neurodegenerative diseases.

Diabetes and idiopathic cardiomyopathy have the same pathological processes as neurodegenerative diseases. Many pieces of research prove that protein aggregates can accumulate in corresponding organs, causing symptoms such as inadequate insulin secretion or cardiac failure [5,34]. In the kidney, some evidence also remind us that protein aggregates can accumulate in renal tubular epithelial cells in a high load and high-pressure state [36]. Also, in chronic kidney disease (CKD) disease models, protein aggregates were significantly increased [104]. Clearing protein aggregates in renal tubular epithelial cells may be an effective way to delay the process of CKD. However, the status of protein aggregates in CKD still needs more research to clarify which role protein aggregates play in CKD.

Interestingly, protein aggregation accumulation also has a good result in a special aging related disease, which is cancer. Some researchers discovered that promoting protein aggregation accumulation can accelerate oncocytes apoptosis and immunogenic cell death [105,106]. Oncocytes were in a hypersynthetic and hypermetabolic state, which cause more sensitivity to protein aggregation. Metabolic processes and the membrane system of oncocytes are disrupted by increasing the load of protein aggregation, So as to achieve the purpose of inhibiting the growth of cancer tissue.

## 7. Conclusions

Protein aggregation is a basic component of cells, the balance of protein aggregation is the key process for cells to keep protein homeostasis. In aging cells, this balance can be broken for lysosome degradation, ER stress, UPS, and so on. Also, protein aggregation accumulation could promote cell aging through breaking protein homeostasis. This sets in motion a self-propagating cycle that exacerbates proteome imbalance and eventually leads to protein homeostasis collapse and disease occurrence. Metabolism of protein aggregation depends on UPP, ALP, ACD, and exophers. Most of them are age-dependent decline, so it can explain why age is the major risk factor for aggregate-deposition diseases, and the nervous and cardiovascular are the generally damaged target systems of protein aggregation. Damage to the protein homeostasis network also injures the cells through a series of interactions. In these interactions, membrane system disruption, DNA damage, protein synthesis environment confusion, and lysosomal function disability are the key links, and they can mutually reinforce. Therefore, regulating protein aggregation is a useful pathway for therapeutic intervention of aging-related diseases and degenerative diseases, even cancer. However, achieving this goal requires a comprehensive understanding of the organization and relationship between protein homeostasis and protein aggregation.

## Figures and Tables

**Figure 1 ijms-24-08593-f001:**
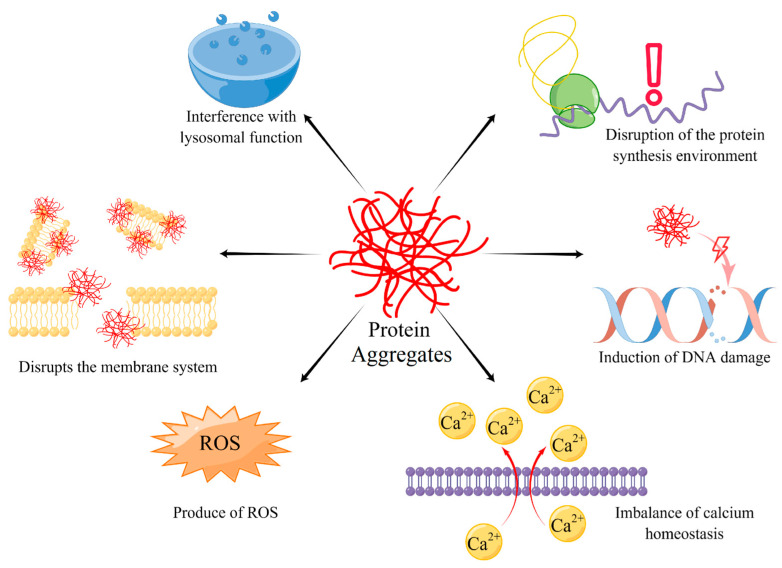
Six main ways of protein aggregates damage the metabolism of cells. They contain interference with lysosomal function, disruption of protein synthesis environment, damage DNA, disturb calcium homeostasis, produce ROS, and injure the membrane system.

**Figure 2 ijms-24-08593-f002:**
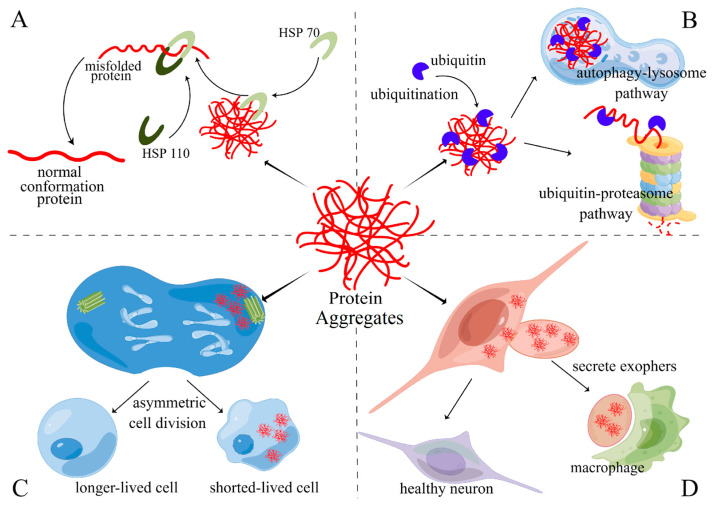
An organism can clear protein aggregates through four pathways: (**A**) HSPs family associate refolding and depolymerizing. In metazoa, HSP70 locate in protein aggregates, and with the help of HSP110, it applies pulling forces to aggregates that disentangle trapped polypeptides; (**B**) ubiquitin-proteasome pathway (UPP) and autophagy-lysosome pathway (ALP) can deal with protein aggregates when their scale exceeds the ability of HSPs family, UUP often degrades soluble misfolded proteins and aggregates, while insoluble proteins or aggregates were disposed of by ALP; (**C**) asymmetric cell division (ACD) can birth a healthier cell with fewer protein aggregates at the cost of another daughter cell with more protein aggregates and shorter life; (**D**) secrete exophers which contain protein aggregates are a useful way for G0 cells such as neurons, exophers would be cleared by macrophages.

## Data Availability

Not applicable.

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
