# Peer review of "Cellular Protein Aggregates: Formation, Biological Effects, and Ways of Elimination"

_ijms, 2023, doi:10.3390/ijms24108593_

Round 1

Reviewer 1 Report (Previous Reviewer 3)

The authors addressed my comments in the first revision. There are some minor points to be considered.

-       I added some comments (related to the English editing) to the PDF file, please find them.

-       For better presentation, the authors must state the problem and its consequences first then the solution. Both “The harm of protein aggregates” and “The function of protein aggregates in development of aging-related diseases “ are suggested to come before “The way to eliminate protein aggregates”.

My comments on the English language are present in the PDF file attached, please find them.

Author Response

Thank you very much for your kind comments and suggestion. We have modified the manuscript according to your suggestions.

Reviewer 2 Report (Previous Reviewer 2)

It is strange the discussion of aggregates does not even consider peptide chain cross-linking in the alpha configuration.  This is why these inclusions cannot be readily digested by proteases which effectively break down peptides in alpha-helices.

English language style is still rather poor

Author Response

Thank you very much for your comments. We did not discuss in detail why proteasome can not degrade protein aggregates, because there are not many studies in this area, and there is no consistent conclusion. On the other hand, this issue is not the focus of this article, so we will not discuss it too much.

Reviewer 3 Report (Previous Reviewer 1)

The authors took into account all the comments of the reviewer and significantly revised the entire text of the article. I believe that now the review can be published as it is.

Significantly improved. Minor editing of English language can be done.

Author Response

Thank you very much for your recognition of our work. And thank you for your comments and suggestions, which make our manuscript further improved.

This manuscript is a resubmission of an earlier submission. The following is a list of the peer review reports and author responses from that submission.

Round 1

Reviewer 1 Report

The review presented by the authors is certainly important, since proteins are involved in all intracellular processes and the violation of their structure entails serious consequences for the cell and the organism as a whole. The aggregation of proteins certainly interferes with their normal functioning. To prevent this process in the cell, there is a network of proteostasis, which includes many components that continuously support a complex protein community. The topic raised by the authors is very broad and multifaceted. The authors have presented a fairly good overview of current works in this broad field. At the same time, it seems to me that some important questions remained unrevealed.

Minor comments:

1. Speaking about protein aggregation, it is necessary to reveal the role of intrinsically disordered proteins and intrinsically disordered regions of globular proteins.

2. It is necessary to reveal the role of LLPS in the formation of protein aggregates, including the formation of amyloid fibrils. There are already many publications on this subject.

3. It will be good to uncover the interaction between biomolecular condensates and proteostasis (see for example Amzallag and Hornstein, 2022).

4. The authors begin the Review with neurodegenerative diseases, and rightly so, but in section 6 they mention them only in passing, for some reason without paying any attention to them.

5. It remains unclear whether the cell has methods of disassembly and removal of mature amyloid fibrils.

6. In conclusion, the authors write "Therefore, regulating of protein aggregation is a useful pathway for therapeutic intervention of aging-relation diseases and degenerative diseases, even cancer" lines 371-372. And what about amyloid fibrils, because the destruction of mature amyloid fibrils is not at all safe?

7. Line 117-118 aggregation of proteins and RNA cannot change the nature of LLPS. Сontrary, сhanging the nature of LLPS can lead to a change in the resulting coacervates. Besides, unnecessary "be" must be removed.

Reviewer 2 Report

This manuscript looks as though it could be valid and interesting if only it were legible. 

However starting with the first sentence: "As the undertaker of life activity, protein need to work in a wide range of environmental and metabolic conditions",  the report is very hard to understand.  An undertaker is a person who buries people!  Almost every line has gross errors of literacy.  The English of this manuscript needs a complete workover by a native English speaker.

This report should be extensively revised before it can be properly reviewed.

Reviewer 3 Report

Authors summarized the causes of protein aggregation, the effect of protein aggregates on the age-related neurodegenerative disorders, and some of the clearance mechanisms of protein aggregates.

Authors didn’t discuss the potential therapeutic strategies targeting protein aggregation for the treatment of age-related neurodegenerative diseases. Authors need to summarize the compounds in different stages of pharmacological development and their protein pathway targets (Ex. https://www.frontiersin.org/articles/10.3389/fnmol.2020.00098/full).

Authors need to add some more information from literature in some sections of the review such as brief explanation of mechanisms of correct protein configuration (in the introduction section).

Authors need to provide some more details about the aggregation of a-synuclein in Parkinson’s disease. The review summarizes only the aggregation of tau and Ab proteins in Alzheimer’s disease.

Information under the section “The association between protein aggregation and aging-relation diseases” is suggested to be under the title “Discussion” because it summarizes the main points discussed in the review.

Figure 1: Please delete “This is a figure” from the title.

Figure 2: Please delete “Based on current research results, we summarized”, the figure should have a title then brief explanation of the figure.

The review needs extensive English editing. Examples:

Line 9: “that caused” should be “that is caused”.

Line 36: “interference biological” should be “interferences biological”.

Line 57: “Organism depend” should be “Organism depends”

Line 344: “aging-relation diseases” should be “aging-related diseases”